# Integrated Ammonia Sensor Using a Telecom Photonic Integrated Circuit and a Hollow Core Fiber

**Andreas Hänsel** [1,*] , **Abubakar Isa Adamu** [2] , **Christos Markos** [2] , **Anders Feilberg** [1] , **Ole Bang** [2] **and Martijn J.R. Heck** [1]

1    Department of Engineering, Aarhus University, 8200 Aarhus, Denmark; af@eng.au.dk (A.F.); mheck@eng.au.dk (M.J.R.H.)
2    DTU Fotonik, Technical University of Denmark, 2800 Kongens Lyngby, Denmark; abisa@fotonik.dtu.dk (A.I.A.); chmar@fotonik.dtu.dk (C.M.); oban@fotonik.dtu.dk (O.B.)
*    Correspondence: a.hansel@eng.au.dk

**Abstract:** We present a fully integrated optical ammonia sensor, based on a photonic integrated circuit (PIC) with a tunable laser source and a hollow-core fiber (HCF) as gas interaction cell. The PIC also contains a photodetector that can be used to record the absorption signal with the same device. The sensor targets an ammonia absorption line at 1522.45 nm, which can be reached with indium phosphide-based telecom compatible PICs. A 1.65-m long HCF is connected on both ends to a single-mode fiber (SMF) with a mechanical splice that allows filling and purging of the fiber within a few minutes. We show the detection of a 5% ammonia gas concentration, as a proof of principle of our sensor and we show the potential to even detect much lower concentrations. This work paves the way towards a low-cost, integrated and portable gas sensor with potential applications in environmental gas sensing.

**Keywords:** Photonic integrated circuit; hollow core fibers; optical gas sensor; indium phosphide

## 1. Introduction

Gaseous emissions from intensive livestock production contribute to global warming and several other environmental challenges. The livestock contribution to global warming is mainly due to emissions of methane and nitrous oxide from animal metabolism as well as manure storage and application. Emission of ammonia from livestock manure contribute to (i) human health effects and premature deaths due to fine particle formation, (ii) eutrophication of aquatic ecosystems, (iii) acidification of soils through nitrification, and (iv) secondary emission of nitrous oxide [1]. Controlling and legislating guidelines on the emission of such gases is hindered by the lack of cost-effective, sensitive and selective gas detection systems. In the case of ammonia, optical gas detectors achieve the required detection and selectivity levels, but are not suitable for wide application due to their size and price [2–5]. Photonic integration targets to reduce the costs and footprint of such devices, such that optical gas detectors can be used outside of research institutions. Recent work has theoretically shown the potential for photonic integrated circuits (PICs) paired with hollow-core fibers (HCFs) as gas sensor for environmental monitoring [6], but in the case of ammonia the experimental verification has not been shown. This paper investigates the absorption lines within reach of telecom foundry processes, leveraging the high technological maturity of indium phosphide (InP) based diode lasers. The combination with HCFs allows for a sensing system with a small footprint, as HCFs allow for long propagation lengths while maintaining small size. Further size reductions can be achieved with a PIC-based gas interaction cell [7]. Tombez et al. demonstrated a silicon photonics methane sensor employing an on-chip interaction cell [8]. However, such cells typically only reach path lengths in the order of ~10 cm, much shorter than what can be achieved with HCFs. These limits

are not only reached due to geometrical constraints on chip, but also due to higher waveguide losses (∼2 dB/cm). The shorter interaction length paired with the lower percentage of the guided mode field in the medium under test (∼30%) require stronger absorption lines for accurate measurements than when using HCFs. A detailed discussion of PIC-based gas sensors regarding laser, interaction cell, and detector can be found in Reference [7]. It is worth mentioning, that the proposed sensor can be adapted for different target gases, such as methane ($CH_4$), carbon monoxide, carbon dioxide ($CO_2$), acetylene, and hydrogen sulfide [6]. All of those gases have absorption lines within the possible emission ranges for telecom photonic integrated circuits. An even wider set of gases can be targeted when relying on other foundry processes that enable optical gas detection with PICs at wavelengths other than the telecom band [7].

## 2. Setup and Components

A schematic of the full setup is shown in Figure 1. It can be separated into three main components: the PIC-based laser source, the HCF gas cell, and the detector. The laser was produced as part of multi-project wafer (MPW) run offered by SMART Photonics. Section 2.1 is dedicated to the description of this PIC. Light-gas interaction takes place in the HCF, which is further expanded on in Section 2.2. The detector is an Agilent 8164B lightwave measurement system (LMWS) with two of its slots being occupied by a 81633A and a 81632A power meter respectively. One of the power meters is used for the measurement path (PM2), the other one for the reference measurement (PM1) to compensate for laser noise and the residual amplitude modulation (RAM) of the tunable laser. Per fiber connector pair, a link loss of ∼0.3 dB can be expected. For the optical path from the chip to PM2, five of such links are encountered, resulting in a loss of ∼1.5 dB. Fiber-chip coupling typically shows a loss of ∼5 dB, and the mechanical fiber splices between the HCF and the SMF have been measured to yield a ∼2.1 dB loss. These values are summarised in Table 1.

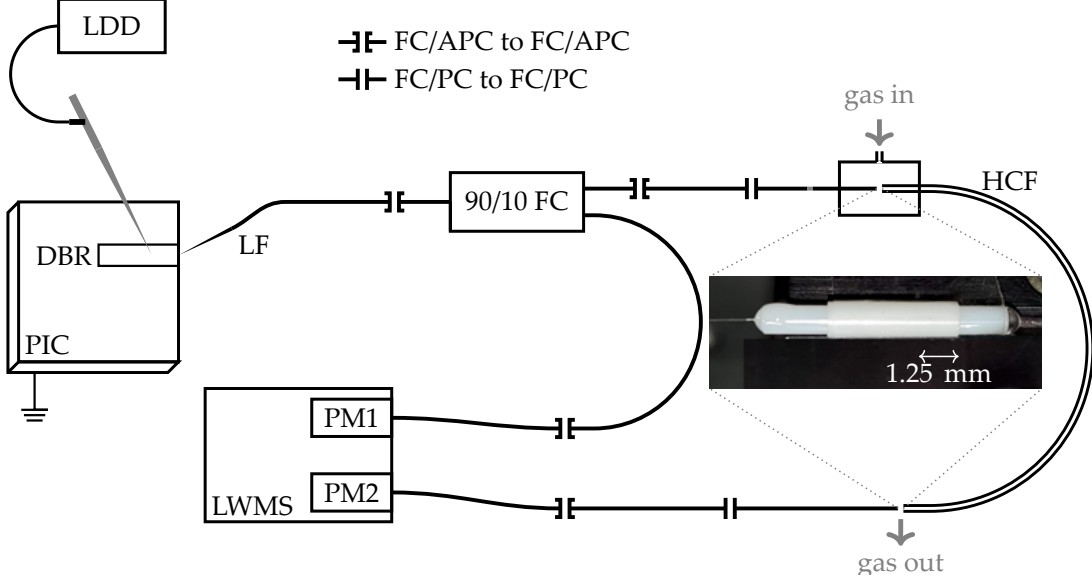

**Figure 1.** Sketch of the setup. The photonic integrated circuit (PIC) is grounded from the substrate side, whereas the laser diode driver (LDD) connects to the on-chip distributed Bragg-reflector (DBR) laser with a probe needle. The laser output is collected with a lensed fiber (LF) and split into two arms at the 90/10 fiber coupler (FC). 10% of the light is guided towards the power meter 1 (PM1) of the light wave measurement system (LWMS). The 90% light path connects to the roughly 1.65 m-long hollow-core fiber (HCF), which has a mechanical splice on each end to allow gas in/out flow. The mechanical splice is shown at the inset of the figure. Light that passed through the HCF is then guided towards the second power meter (PM2).

### 2.1. PIC

The PIC is an indium phosphide based distributed Bragg reflector (DBR) laser. There are two DBR gratings of equal length (200 μm) and pitch, which sandwich a semiconductor optical amplifier (SOA; 250 μm length). Both DBR and SOA can be contacted by needle probing, but for the tuning operation that is needed here, only contacting the SOA suffices. The ground contact is established on the substrate side of the chip. The temperature of the chip is controlled with a thermoelectric cooler and a thermistor. Figure 2 shows the corresponding mask as submitted to the foundry (SMART Photonics), as well as the small segment of the DBR laser with the landed needled and lensed fiber. The chip contains other laser structures designed for different wavelengths, that can be used to target different absorption lines, as well as on-chip photodetectors (PDs). The chip is $2 \times 4.6\,\text{mm}^2$ in size. Earlier work has shown, that a widely tunable laser can be realised on the same platform, but would require a more complex tuning algorithm [9–11]. The DBRs have been designed to have the center of their reflection at a wavelength of 1514 nm, but due to manufacturing uncertainties, the eventual maximum was shifted to longer wavelengths, which allowed for targeting the 1522.45 nm absorption line of ammonia. This absorption line, as calculated with HITRAN [12] for a length of 1.65 m of a gas mixture consisting of 5% pure ammonia at atmospheric pressure, is shown in Figure 2b. As can be seen, the full width at half-maximum (FWHM) of the absorption line is about 0.1 nm, so less than 0.5 nm of tuning range are sufficient for a measurement on this line. The linewidth of such PIC lasers is typically limited by the laser diode driver, and is typically below ~1 MHz [13], i.e., small in comparison to the ammonia absorption linewidth (~9 GHz). The combination of SOAs and PDs on the same circuit allows for several advanced spectroscopic configurations, such as having the reference arm on chip, or balanced detection schemes. Wavelength modulation spectroscopy (WMS) has been recently demonstrated in a HCF-based $CO_2$ and $CH_4$ sensor [14].

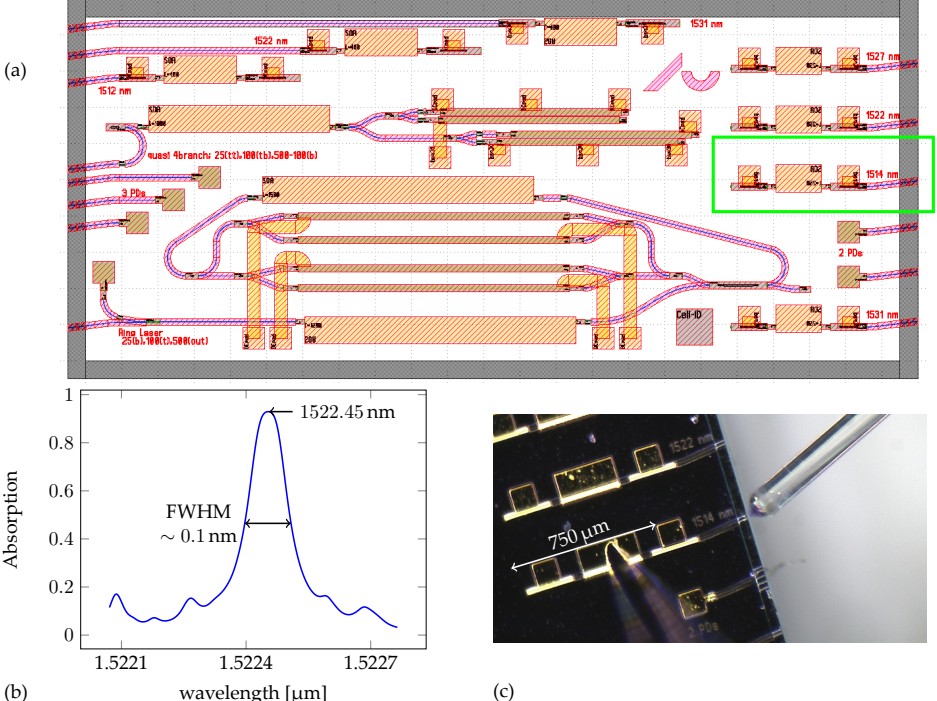

**Figure 2.** (**a**) Mask submitted to the foundry. The laser used in the experiment is framed in green. The chip is $2 \times 4.6\,\text{mm}^2$ in size. (**b**) Absorption of 5% ammonia gas mixture after 1.65 m propagation at atmospheric pressure. (**c**) Microscope image of the laser structure as highlighted in green in (**a**). The picture shows the probe needle landed on the metal pad to inject current to the semiconductor optical amplifier (SOA) as well as the lensed fiber used for coupling.

*2.2. HCF*

The HCF used is a commercially available hollow core photonic band gap (HCPBG) fiber (NKT Photonics 7-cell HC-1550-2). The 11 μm core diameter HCPBG fiber has a loss of <30 dB/km within the transmission window (1490 nm to 1680 nm) with extremely low bend loss, which means that a bend radius of less than 1 cm can be used without perturbations to the guided light [15]. In devising the all-fiber system, we deployed a mechanical splice at the two ends of the HCPBG fiber similar to that reported is [16–18]; where the ends of the HCPBG fiber are mated with a ceramic mating sleeve to a single mode fiber (SMF; inset Figure 1), after both facets have been cleaved with a tension cleaver. Using an optical microscope, optimum alignment between the SMF and the HCPBG fiber is ensured, resulting in a gap of ∼50 μm, with ∼2.1 dB insertion loss at each junction of the mechanical splice (including Fresnel reflection at ends of SMF). The gap between the fibers functions as an inlet/outlet for the gas sample. The beam-profile is also measured at each junction during the mechanical splice to ensure that light is guided in the fundamental mode of the fiber. The beam profile at the exit of the HCF as measured with a Thorlabs BP209-IR2/M is shown in Figure 3a. The fiber length is important in the performance of the gas sensor, as the absorption is directly proportional to the fiber length (as well as the concentration and line strength), as described by the Beer-Lambert law. The transmission of light through the gas sample is then

$$I = I_0 \exp(-\alpha) \tag{1}$$
$$\alpha = S(T)g(\nu, T, p)nl, \tag{2}$$

where $I$ and $I_0$ denote the transmitted and incoming intensity, and $\alpha$ the absorption coefficient [19]. The absorption coefficient depends on the (temperature-dependent) line strength $S(T)$, the normalised lineshape $g$, the absorber number density $n$ and the path length in the absorbing medium. The lineshape is a function of the light frequency $\nu$, the temperature $T$, and the pressure $p$. As can be seen, for a given detectable absorption strength, the concentration can be decreased if the path length is increased accordingly. It is also worth noting, that while there is a temperature dependence, the signal changes very slightly with temperature, and its impact on the absorption coefficient can be ignored in the current setting [20]. Increasing the pressure leads to an increased amount of absorbers in a given volume, and hence a stronger absorption; however, the increased pressure also results in stronger collision broadening, leading only to slight changes when comparing the maximum absorption level to the baseline. Operating the device at higher pressure can be beneficial when using different methods, e.g., integrative measurements or fitting routines, to determine the concentration. For this experiment, a length of ∼1.65 m is deployed. While a longer length will yield better results, i.e., a higher absorption, there is a trade-off as the longer length will require longer filling time, which consequently increases the sensor response time. Therefore, increasing the length of fiber would allow for detecting lower concentrations or targeting weaker absorption lines, e.g., from other gas species. For faster filling of the fiber the pressure was increased to ∼1.5 bar, but the pressure was reduced to atmospheric pressure when taking the data. The response times for different pressures have been presented in an earlier study [17]. While the low fiber loss of <30 dB/km would permit the use of a much longer fiber, the length of ∼1.65 m was chosen as a balance between absorption strength and filling time, which is in the order of a few minutes [15].

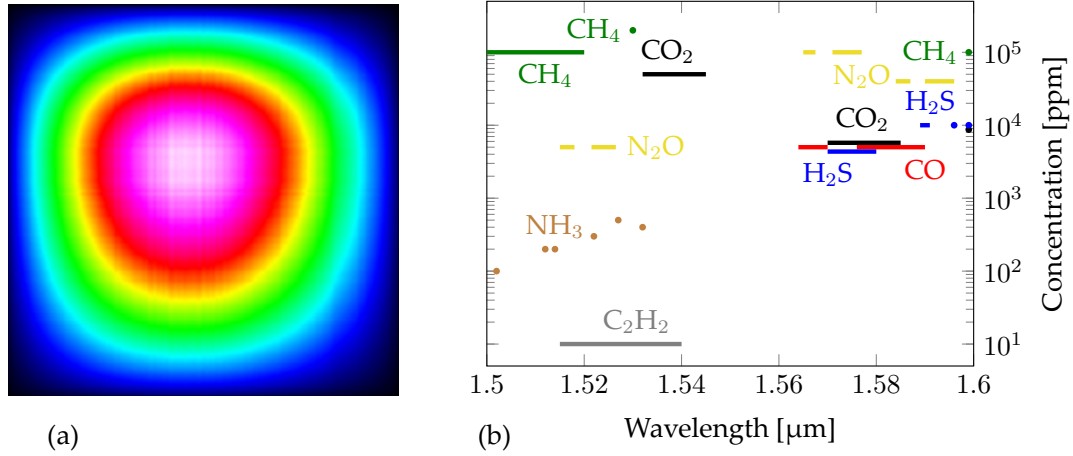

**Figure 3.** (**a**) Beam profile measured after the output of the hollow core fiber. It can be seen that the fundamental mode is the predominant mode transmitted through the fiber. (**b**) Detectable concentrations of several gases that have absorption lines in the range exploitable by telecom technology photonic integrated circuits. The shown concentrations correspond to signal strengths of ~0.1% and assume a propagation length of 1 m in neutral background at atmospheric pressure at room temperature. Lines in the diagram represent a close set of multiple usable absorption lines, whereas a dot refers to a single line.

**Table 1.** Estimated link loss per transition element for the main path in the setup, i.e., from the on-chip distributed Bragg-reflector (DBR) laser to PM1 on the light wave measurement system (LWMS), as shown in Figure 1.

| Element | Loss [dB] | Occurence | Total [dB] |
|---|---|---|---|
| FC/PC-FC/PC | 0.3 | 2 | 0.6 |
| FC/APC-FC/APC | 0.3 | 3 | 0.9 |
| SMF-HCF | 2.1 | 2 | 4.2 |
| chip-fiber | 5 | 1 | 5 |
| 90/10 FC excess loss | 0.15 | 1 | 0.15 |
| Total | | | 10.85 |

## 3. Results

The DBR laser was driven at a temperature of 26 °C to tune laser emission to the desired wavelengths. At this temperature, the threshold current was approximately 17 mA and the laser could reach a maximum output power of ~2 mW. The side mode suppression ratio (SMSR) was measured with an Yenista/EXFO OSA20 and exceeded 30 dB. The laser emission wavelength can be tuned by changing the driving current through the SOA, which was used for the fine tuning to sweep over the absorption line. This leads to a RAM, which can be compensated for by using a reference detector, i.e., PM1 in our setup. Figure 4a shows the reference power as measured with PM1, as well as the transmitted power through the HCF filled with a mixture of 5% ammonia and 95% nitrogen as neutral background gas. As can be seen from the figure, increasing the injection current into the SOA increases the measured reference powers. This is due to an increase in emitted laser light, which is detected by PM1. Current injection into the SOA is also used to tune the emission wavelength of the laser, as the increased current affects the electric carriers in the semiconductor, as well as the temperature. Temperature and carrier density affect the refractive index of the waveguiding material, and hence the effective index. This change in index affects the longitudinal mode of the laser cavity. The increase in emitted laser light power when tuning the laser wavelength, as seen in Figure 4a in red, requires normalisation when scanning over an absorption line, e.g., with a reference detector, and is called RAM. This is also seen when following the blue line, as a slight increase of the base line power,

i.e., the measured power without the presence of a strong gas absorption for a certain wavelength, is visible. In Figure 4a this can be seen when following the blue line: between 50 and 60 mA, the power only slowly increases. There are faster variations on that curve, that are attributed to noise as well as side absorption lines of much lower strength than the main one. The main absorption line can be observed between the ∼63 mA and 73 mA where the curve seems to to show a dip. Here, dip refers to the segment at the curve where the transmitted power falls from ∼0.3 mW to a level that is ∼90% lower than that (∼0.03 mW). For injection currents in excess of 73 mA, a small increase in the measured power, similar to the section of currents lower than 63 mA, is observed. By normalising the transmitted power with the reference power, the absorption line can be quantified. The result of this normalisation, overlapped with the expected transmission signal predicted by HITRAN is shown in Figure 4b. Figure 4b shows the absorption line as calculated with HITRAN in red. The line contains one very strong absorption line in the center, as well as sidelines at much lower absorption strengths. For lower driving currents, the shape of the absorption as shown in red can be quite accurately predicted, i.e., comparing the shape of the blue and the red line, a nice overlap can be observed. As blue shows the measured data of a normalized power, which can be seen as equivalent to a transmission measurement, and red the predicted transmission, a good correspondence between measurement and theory can be attested. The blue curve shows seemingly random fluctuations that distort the quality of the curve, which can be attributed to the noise of the system. At higher currents the tuning seems to stagnate, i.e., the emission wavelength of the laser seems to change less for the same current change when compared to lower injections currents. This can be seen when following the normalized power of Figure 4b beyond 70 mA, where some of the absorption line features given by HITRAN can be seen, but at the wrong position with respect to the x-axis. Figure 4b also allows for an estimate of the detection limits of the system. From the blue line of Figure 4b, noise fluctuations of the normalized transmission are at the 10% -level peak to peak. Similarly the modulations (i.e., the change of the measured power outside of noise fluctuations, between 50 and 70 mA) allow the observation of features that are expected when comparing to the HITRAN transmission, which are also in the range of 10% . The signal to noise ratio (SNR) can be potentially improved by averaging. The data points in Figure 4 have been taken with an integration time of 0.1 s. However, beyond averaging times of 1 s, no further noise reduction could be seen. This is attributed to the interaction of different spatial modes in the HCF [14,21]. Overall, a sampling of 0.1 s provided the best trade-off between number of data points, and hence proper sampling of the absorption line, and noise. Due to fluctuations from mechanical drifts, it is important to complete a measurement in a short window, This window can range from 10 minutes to 2 hours. This influence from lab conditions can be eliminated by integrating into a fully packaged system.

Figure 5 shows the measurement data when using the on-chip photodetector instead of an external device. The light output that would lead to the external power meter PM2 in Figure 1 was coupled to a PD on the opposite side of the PIC. The photo current was measured for a bias voltage of −2 V. The measured photocurrent can be related with the power of light impinging the detector via the responsivity, which is typically around 0.85 A/W for PDs of this foundry [22]. Taking this estimated conversion value, a photocurrent of 5 µA corresponds to a power of 4.25 µW. However, the exact conversion ratio depends on a multitude of parameters, among others, wavelength, bias voltage, power levels, and temperature. Nevertheless, at least for a small interval, a linear relation between power and measured photocurrent can be assumed. Figure 5a shows an increase in the emitted laser power when increasing the SOA current. This has been identified before as RAM and is a consequence of the tuning mechanism of the laser. As the laser is unchanged, the same RAM features can be observed, although the detector has changed. Comparing the blue and the red curves in Figure 5a, the blue curve has a small segment between 60 and 80 mA, where the measured photocurrent falls significantly faster than the reference power. This is attributed to a dip in transmission due to laser light absorption by ammonia in the HCF. Due to the underlying RAM, when not normalising the measured photocurrent, this gives the impression of a stronger dip following the curve from the left.

This asymmetry is a consequence of the continuously increasing laser emission power. Despite the fixed gas composition for the measurements shown in Figures 4 and 5, the measured absorption signal, i.e., the dip in the measured photocurrent, decreased to 10% of its maximum value in Figure 5a (from ~4.8 to ~4.4 µA). Assuming the typical responsivity values for this platform, this translates into ~4.1 and ~3.7 µW. The dip of 10% in Figure 5a is lower than the dip of 90% observed in Figure 4. This can be expected, as no polarisation control has been included in the setup and the waveguides. Meanwhile, the photodetectors are very polarisation sensitive. Thus, light from the backside of the DBR laser can reach the PD, although no waveguide has been etched to link them. This can be verified from Figure 5b, where the lensed fibers have been intentionally misaligned to prevent effective fiber-chip coupling. Despite the lack of this coupling, the measured photocurrent increases with growing injection current to the SOA (blue). The red curve shows that even with the bad alignment, lasing can be verified with the external reference detector PM1. Since a big part of the light that is measured by the on-chip photodetector did not propagate through the HCF, and hence never interacted with the ammonia within said fiber, a large fraction of the measured power does not contain any absorption signal. Despite these unfavourable conditions, an absorption dip can be identified.

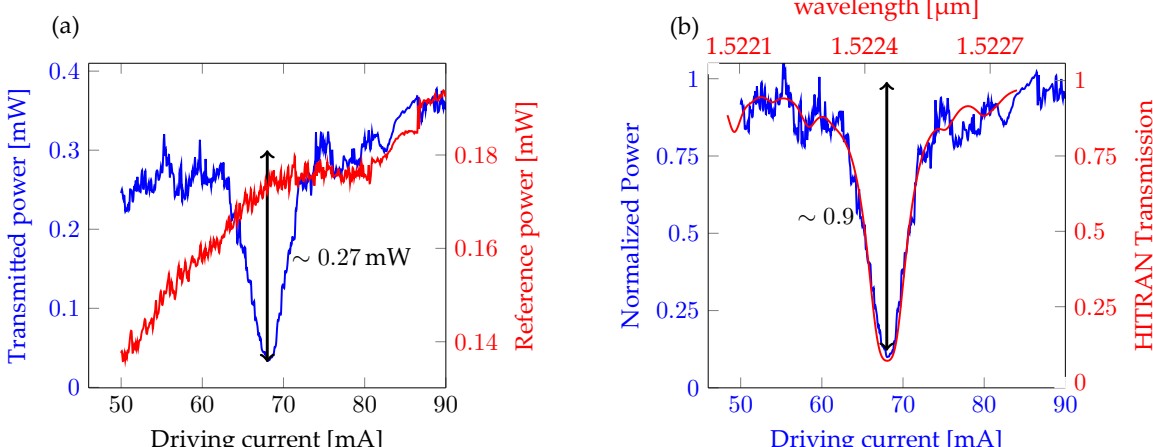

**Figure 4.** (**a**) Blue: Measured power through the filled hollow-core fiber (HCF) (5% ammonia). Red: Measured power on PM1. (**b**) Blue: Normalised power as a function of driving current. Red: expected transmission according to HITRAN as a function of wavelength.

These measurements shows the first fully integrated ammonia measurement using a PIC and a HCF. The sensor can easily be adapted to target other gases that show absorption in the near-infrared wavelengths. The measurement can be improved by placing absorbers, such as another PD, between the DBR laser and the measurement PD, which eliminates the erroneous signal, and incorporating a proper polarisation control for the coupling to the PD, which will effectively increase the signal strength. Future designs should also facilitate asymmetric DBR laser structures. As those measures increase the amount of signal that reaches the detector without a significant change to laser and detector noise, the SNR of should increase. The bottleneck of the measurements using the external photodetector is the modal interference in the HCF. The improvements of detector and laser should allow for SNRs comparable to the measurements with the external photodetector, as only the HCF is unchanged and the predicted improvements of the other components go beyond the SNR limits set by the HCF. Figure 6 shows the stability of the HCF when spliced to a SMF on each end when using a broadband light source, in this case with a FWHM of 10 nm. Due to the broad illumination, modal interference is effectively eliminated, and much lower noise levels can be reached ($\sim \frac{0.6\,\mathrm{mW}}{30\,\mathrm{mW}} = 2\%$ peak-to-peak).

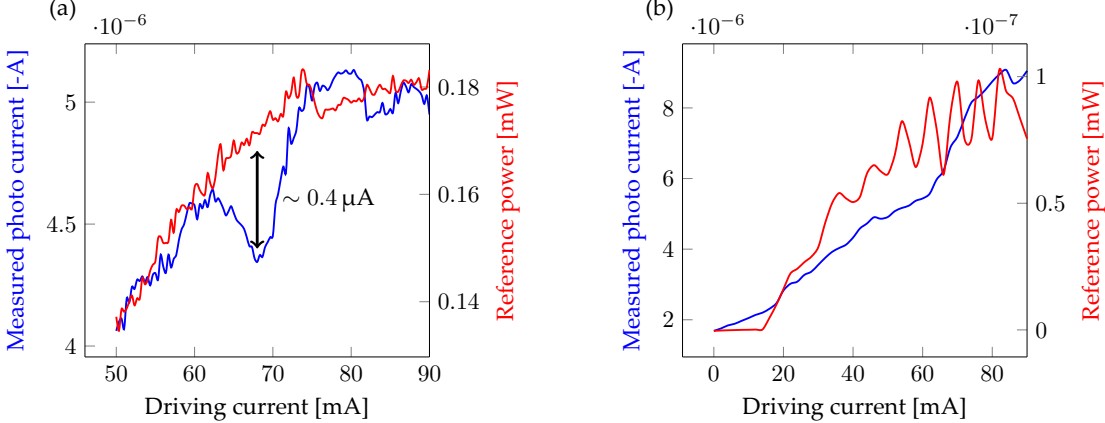

**Figure 5.** (**a**) blue: Current on on-chip photodetector (PD) red: reference power. (**b**) blue: Current on on-chip PD without fiber coupling. red: Amplitude rise on reference due to coupling into unaligned fiber on laser side. In contrast to other measurements, these curves have been taken at 18 °C.

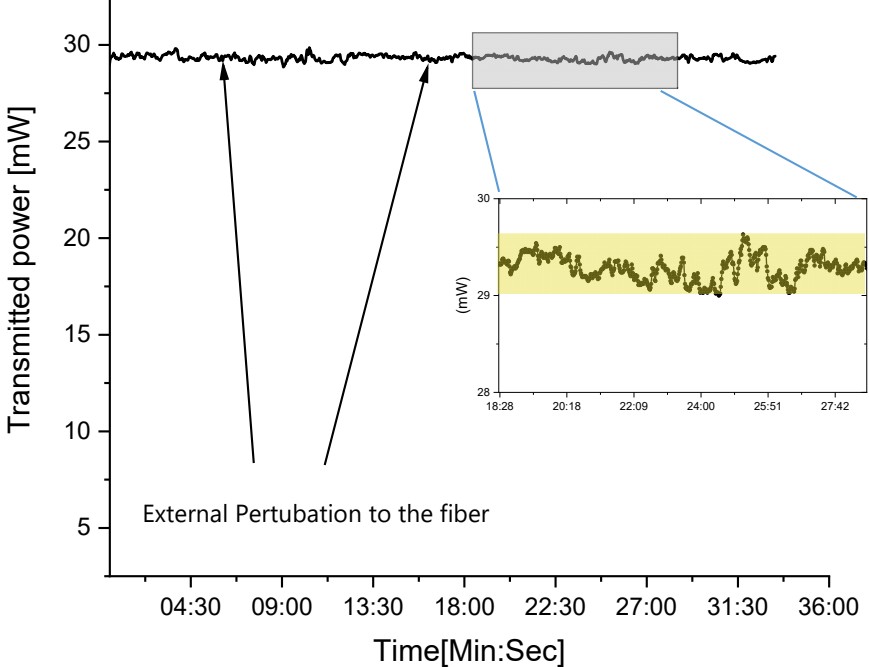

**Figure 6.** Long time measurement of the stability of the hollow-core fiber including the mechanical splices to the single mode fibers on each end. The inset shows a zoom in on a shorter time frame, showing fluctuations at the 2% level. For these measurements a broadband light source has been used that effectively eliminated the noise originating from modal interference.

## 4. Conclusions

We presented the first ammonia measurement using a PIC and a HCF. The PIC was relying on mature telecom foundry processes targeting absorption wavelengths in the near-infrared. For verification purposes we also investigated the sensitivity of the system with an external reference. Currently absorptions of 10% power, i.e., the power in the absorption line drops by 10% of its baseline value, can be detected. Using the HITRAN database values for linestrengths, absorption strengths can be linked to concentrations, deduced from the Beer-Lambert law. When using the absorption line at ∼1522.45 nm, an absorption strength of 10% corresponds to an ammonia concentration of

∼0.1% [12]. However, the system can be easily adapted for other target wavelengths and hence gases, e.g., methane. An overview over a few gases that can be detected using this technology is shown in Figure 3b. The shown concentrations correspond to a signal strength of ∼0.1% and a propagation of 1 m at atmospheric pressure and room temperature. The corresponding wavelengths of those absorption lines can be reached by changing the pitch of the DBR gratings that were used in this design, or by relying on widely tunable laser (WTL) structures. Several WTL concepts have already been demonstrated within the same platform [9–11]. Photonic integration also lends itself to more advanced detection methods, such as wavelength modulation spectroscopy (WMS) or balanced detection methods. For an increase in stability and a portable solution, PICs can be packaged.

**Author Contributions:** Project administration and funding acquisition, A.F., O.B. and M.J.R.H.; conceptualisation, C.M. and M.J.R.H.; formal analysis, investigation and visualization, A.H. and A.I.A.; writing–original draft preparation, A.H. and A.I.A.; writing—review and editing, all authors. All authors have read and agreed to the published version of the manuscript.

**Funding:** This work is funded by the Innovation Fund Denmark (project number 6150-00030B).

**Conflicts of Interest:** The authors declare no conflict of interest.

## Abbreviations

The following abbreviations are used in this manuscript:

| | |
|---|---|
| PIC | photonic integrated circuit |
| HCF | hollow-core fiber |
| SMF | single-mode fiber |
| LMWS | lightwave measurement system |
| DBR | distributed Bragg reflector |
| SOA | semiconductor optical amplifier |
| PD | photodetector |
| FWHM | full width at half-maximum |
| HCPBG | hollow core photonic band gap |
| SMSR | side mode suppression ratio |
| RAM | residual amplitude modulation |
| SNR | signal to noise ratio |
| WTL | widely tunable laser |

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

**Publisher's Note**: MDPI stays neutral with regard to jurisdictional claims in published maps and institutional affiliations.

