# Peer review of "Integrated Ammonia Sensor Using a Telecom Photonic Integrated Circuit and a Hollow Core Fiber"

_photonics, doi:10.3390/photonics7040093_

Round 1

Reviewer 1 Report

In this paper, Authors present a fully integrated optical ammonia sensor based on a photonic integrated circuit (PIC) with a tunable laser source and a hollow-core fiber (HCF). Detection of a 5% ammonia gas concentration is demonstrated.

I would recommend the following revisions before manuscript publication:

1) A detailed description of the setup including power penalties at different connection / coupling points is reported in the manuscript. Authors should include a summary table with detail description of the link budget analysis (from optical source to detectors, i.e. on-chip PDs and/or PM). In this way, a reader would be able to build up the sensor system probably using different components, but having a clear definition of the power spec. needed to detect x% of ammonia (or other gases included in Fig. 3b).

2) Authors should report a more detailed analysis (design plot) of the length of the HCF as a function of optical signal attenuation and gas absorption efficiency. Selection of 1.65m-long HCF as optimal trade-off in this experiment should be demonstrated.

3) Can Authors compare performance achieved by their sensing system with other technology platforms ? For example, an SOI or SiN high-Q ring resonator sensors on-chip (without using an external HCF). This comparison (pros & cons) with state-of-art PIC sensors might be also useful for improving the Introduction Paragraph.

Author Response

Dear Reviewer,

Thank you for the positive feedback and constructive recommendations regarding the manuscript. We believe this has helped improve the clarity and quality of the paper. In this letter, we will respond to the points mentioned in the review report and indicate which changes have been made to manuscript in accordance with those points. Quoted text will be in italic font; text added to the manuscript will be placed in red (and highlighted in the PDF-version of the manuscript).

In this paper, Authors present a fully integrated optical ammonia sensor based on a photonic integrated circuit (PIC) with a tunable laser source and a hollow-core fiber (HCF). Detection of a 5% ammonia gas concentration is demonstrated.

I would recommend the following revisions before manuscript publication:

  • 1) A detailed description of the setup including power penalties at different connection / coupling points is reported in the manuscript. Authors should include a summary table with detail description of the link budget analysis (from optical source to detectors, i.e. on-chip PDs and/or PM). In this way, a reader would be able to build up the sensor system probably using different components, but having a clear definition of the power spec. needed to detect x% of ammonia (or other gases included in Fig. 3b).

Response: We agree with the reviewers comment that the details of setup be provided. To this end, we have added this information to the setup section of the manuscript. We also included a table with the link budget for the main path in the same section. Other optical paths can be easily obtained with these data.

Per fiber connector pair, a link loss of 0.3dB can be expected. For the optical path from the chip to PM2, five of such links are encountered, resulting in a loss of 1.5dB. Fiber-chip coupling typically shows a loss of 5dB, and the mechanical fiber splices between the HCF and the SMF have been measured to yield a 2.1dB loss. These values are summarised in Table 1.

  • 2) Authors should report a more detailed analysis (design plot) of the length of the HCF as a function of optical signal attenuation and gas absorption efficiency. Selection of 1.65m-long HCF as optimal trade-off in this experiment should be demonstrated.

Response: The influence of the fiber length on the signal attenuation is described by the Beer-Lambert law. We expanded the manuscript with a discussion of this law and its consequences on our system. The addition reads:

The transmission of light through the gas sample is then

I = I_0\exp(-\alpha)

\alpha = S(T)g(\nu, T, p)nl

where I and I_0 denote the transmitted and incoming intensity and \alpha the absorption coefficient [19]. The absorption coefficient depends on the (temperature-dependent) line strength S(T), the normalised lineshape g, the absorber number density n and the path length in the absorbing medium. The lineshape is a function of the light frequency \nu, the temperature T, and the pressure p. As can be seen, for a given detectable absorption strength, the concentration can be decreased if the path length is increased accordingly. It is also worth noting, that while there is a temperature dependence, the signal changes very slightly with temperature, and its impact on the absorption coefficient can be ignored in the current setting [20].

Two references have been added:

[19] Hieta, T.; Merimaa, M. Spectroscopic Measurement of Air Temperature. International Journal of Thermophysics 2010, 31, 1710–1718. doi:10.1007/s10765-010-0833-6.

[20] Hänsel, A.; Reyes-Reyes, A.; Persijn, S.T.; Urbach, H.P.; Bhattacharya, N. Temperature measurement using frequency comb absorption spectroscopy of CO2. Review of Scientific Instruments 2017, 88, 053113, doi:10.1063/1.4984252.

The attenuation of the optical fiber is low (<30 dB/km), which allows for very long path lengths. Defining the optimum fiber length, however, is not only based on the absorption and attenuation ratio, but also on the filling time of the fiber. In our case, the measurements have to be undertaken in the order of a few minutes, limiting the length of the fiber. We added the following sentence to clarify that point:

While the low fiber loss of <30 dB/km would permit the use of a much longer fiber, the length of 1.65 m was chosen as a balance between absorption strength and filling time, which is in the order of a few minutes [15].

  • 3) Can Authors compare performance achieved by their sensing system with other technology platforms ? For example, an SOI or SiN high-Q ring resonator sensors on-chip (without using an external HCF). This comparison (pros & cons) with state-of-art PIC sensors might be also useful for improving the Introduction Paragraph.

Response: We agree with the reviewer and have included a more detailed discussion of those systems. On-chip gas cells are an interesting lane of research, but are not available on the indium phosphide platform and hence cannot be monolithically integrated; as such the interaction cell would still have to be considered “external”. We added the following sentences to the introduction to explain our focus on HCFs as interaction cells:

Further size reductions can be achieved with a PIC-based gas interaction cell [7]. Tombez et al. demonstrated a silicon photonics methane sensor employing an on-chip interaction cell [8]. However, such cells typically only reach path lengths in the order of 10 cm, much shorter than what can be achieved with HCFs. These limits are not only reached due to geometrical constraints on chip, but also due to higher waveguide losses (2 dB/cm). The shorter interaction length paired with the lower percentage of the guided mode field in the medium under test (30%) require stronger absorption lines for accurate measurements than when using HCFs. A detailed discussion of PIC-based gas sensors regarding laser, interaction cell, and detector can be found in Reference [7].

Reference [8] was newly added to the manuscript:

[8] Tombez, L.; Zhang, E.J.; Orcutt, J.S.; Kamlapurkar, S.; Green, W.M.J. Methane absorption spectroscopy on a silicon photonic chip. Optica 2017, 4, 1322–1325. doi:10.1364/OPTICA.4.001322.

We want to thank the reviewer once more for the swift comments to improve the manuscript and hope we have incorporated the suggestions adequately.

Reviewer 2 Report

The article proposed for review is devoted to the study of the effect of ammonia on the spectrum of radiation transmitted through it. This is a superb experimental work done. In my opinion, the authors of the experimental work were able to solve a number of technical difficulties. Among which I would like to note: 1. the task of joining a standard telecommunication fiber with an HCF fiber; 2. the task of introducing a gas mixture into the HCF fiber; and the technical task of organizing the reference optic channel. On the basis of a well-organized experiment, the authors were able to confirm the presence of an ammonia absorption line at a wavelength of 1522.45 nm.

At the same time, I have a number of questions for the authors.

  1. It is known that the absorption line of ammonia is at a wavelength of 1522.45 nm. It is known that if laser radiation at this wavelength is passed through ammonia, then part of the light will be absorbed and at the output and the intensity of the light flux will decrease. Where exactly do the authors see the scientific novelty of their work?
  2. Does it make sense to dwell in such detail on the design and principle of operation of the laser? First, the design and operation of the laser are not the subjects of the authors' research work. Secondly, the laser is used only as a tunable radiation source, which can emit at a wavelength of 1522.45 nm.
  3. Is there a need to tune the laser radiation to construct the spectral response?
  4. I understand that the authors used a laser with a narrow emission line (50 kHz) to restore the spectral characteristics of the absorption line. Does it make sense to choose a laser with a center wavelength and a spectral linewidth equivalent to the absorption spectral width of ammonia? In this case, the dependence of the intensity of the absorbed light on the concentration of ammonia could be more pronounced.

In my opinion, the work would greatly benefit if the authors included in the research the influence of the length of the sensor on the measurement results. Would suggest calibration characteristics, for example, the dependence of the influence of the length of the sensor on the absorption coefficient and the dependence of the absorption coefficient on the concentration of ammonia. The authors did not investigate the influence of the sensor temperature on the measurement results. These results could have embellished the work.

I would like to recommend this article for publication after I see it researching the dependence of sensor readings on HCF fiber sensor length, gas concentration, and sensor temperature.

The work doesn't look complete without them.

Author Response

Dear Reviewer,

Thank you for the swift feedback regarding the manuscript. In this letter, we will respond to the points mentioned in the review report and indicate which changes have been made to manuscript in accordance with those points. Quoted text will be in italic font; text added to the manuscript will be placed in red (and highlighted in the PDF-version of the manuscript).

The article proposed for review is devoted to the study of the effect of ammonia on the spectrum of radiation transmitted through it. This is a superb experimental work done. In my opinion, the authors of the experimental work were able to solve a number of technical difficulties. Among which I would like to note: 1. the task of joining a standard telecommunication fiber with an HCF fiber; 2. the task of introducing a gas mixture into the HCF fiber; and the technical task of organizing the reference optic channel. On the basis of a well-organized experiment, the authors were able to confirm the presence of an ammonia absorption line at a wavelength of 1522.45 nm.

Thank you for the very positive assessment and acknowledgement of the work reported in the manuscript.

At the same time, I have a number of questions for the authors.

  • It is known that the absorption line of ammonia is at a wavelength of 1522.45 nm. It is known that if laser radiation at this wavelength is passed through ammonia, then part of the light will be absorbed and at the output and the intensity of the light flux will decrease. Where exactly do the authors see the scientific novelty of their work?

Response: The novelty lies in the integration of commercially available technology to a portable and stable gas sensor system. Optical gas sensors have proven their performance in other settings, but the question remained what kind of sensitivities can be achieved when relying on telecom PIC technology (typically ammonia spectroscopy is undertaken at longer wavelengths, where stronger absorption lines can be found) and HCFs (allow for a long path length in a small volume; no cavity alignment needed). The presented approach paves the way to a gas sensor that is cheap and robust enough to find wide spread application in environmental gas sensing. The novelty and impact is found in the combination of the components for this application.

  • Does it make sense to dwell in such detail on the design and principle of operation of the laser? First, the design and operation of the laser are not the subjects of the authors' research work. Secondly, the laser is used only as a tunable radiation source, which can emit at a wavelength of 1522.45 nm.

Response: We thank the reviewer for this comment. We do see great value in these details, as the laser cannot be bought as a “off the shelf”-component, but reproducing the results requires the submission of the design to the used commercial foundry. We specifically designed the laser for this application. In addition to that, it gives an idea about the size of the device and the scalability, as a single wafer can produce multiple of such laser structures. Lastly, it also highlights the ease of adapting the design to target a different gas species, as well as ways of optimising the design to eliminate the identified flaws in this chip layout.

  • Is there a need to tune the laser radiation to construct the spectral response?

While not per se necessary, tuning the laser over the absorption line allows to obtain the base line, which can be used to calculate the absorption line strength with almost no calibration, as well as to ensure that the laser emission wavelength does not drift.

  • I understand that the authors used a laser with a narrow emission line (50 kHz) to restore the spectral characteristics of the absorption line. Does it make sense to choose a laser with a center wavelength and a spectral linewidth equivalent to the absorption spectral width of ammonia? In this case, the dependence of the intensity of the absorbed light on the concentration of ammonia could be more pronounced.

Response: As long as the laser linewidth is small in comparison to the absorption linewidth, the reading of the line strength is simple. If the laser line gets broader, however, a significant amount of the light can fall outside of the absorption line and hence contribute to the signal at a lesser degree; resulting in a weaker absorption signal, in comparison to the case where all the light is directly at the absorption peak.

To avoid misunderstandings, we removed the line from the results section that mentioned the linewidth (50 kHz), as it is narrower than we expected, and could give the impression that such a linewidth is needed for the measurements. However, any linewidth that is small to the ammonia absorption linewidth (~GHz) suffices.

  • In my opinion, the work would greatly benefit if the authors included in the research the influence of the length of the sensor on the measurement results. Would suggest calibration characteristics, for example, the dependence of the influence of the length of the sensor on the absorption coefficient and the dependence of the absorption coefficient on the concentration of ammonia. The authors did not investigate the influence of the sensor temperature on the measurement results. These results could have embellished the work.

Response: The dependence of concentration, path length and temperature on the absorption strength is adequately described by the Beer-Lambert law. We expanded the manuscript to include a description of said law. The addition reads:

The transmission of light through the gas sample is then

I = I_0\exp(-\alpha)

\alpha = S(T)g(\nu, T, p)nl

where I and I_0 denote the transmitted and incoming intensity and \alpha the absorption coefficient [19]. The absorption coefficient depends on the (temperature-dependent) line strength S(T), the normalised lineshape g, the absorber number density n and the path length in the absorbing medium. The lineshape is a function of the light frequency \nu, the temperature T, and the pressure p. As can be seen, for a given detectable absorption strength, the concentration can be decreased if the path length is increased accordingly. It is also worth noting, that while there is a temperature dependence, the signal changes very slightly with temperature, and its impact on the absorption coefficient can be ignored in the current setting [20].

Two references have been added:

[19] Hieta, T.; Merimaa, M. Spectroscopic Measurement of Air Temperature. International Journal of Thermophysics 2010, 31, 1710–1718. doi:10.1007/s10765-010-0833-6.

[20] Hänsel, A.; Reyes-Reyes, A.; Persijn, S.T.; Urbach, H.P.; Bhattacharya, N. Temperature measurement using frequency comb absorption spectroscopy of CO2. Review of Scientific Instruments 2017, 88, 053113, doi:10.1063/1.4984252.

  • I would like to recommend this article for publication after I see it researching the dependence of sensor readings on HCF fiber sensor length, gas concentration, and sensor temperature. The work doesn't look complete without them.

Response: Similar to the comment before, the Beer-Lambert covers to overall relation and, e.g., how an increase in length allows for lower gas concentrations with the same absorption coefficient. Temperature effects on the absorption line strength are negligible, and as presented here the chip needs to be kept stable at 26 degree Celsius, as otherwise the laser does not emit at the right wavelength. The laser was not designed to be used at temperature much lower than room temperature, where other problems (e.g. water condensation) can occur. We added the following lines to indicate how the temperature stabilization has been achieved:

The ground contact is established on the substrate side of the chip. The temperature of the chip is controlled with a thermoelectric cooler and a thermistor.

We want to thank the reviewer once more for the swift comments to improve the manuscript and hope we have incorporated the suggestions adequately.

Round 2

Reviewer 2 Report

Dear Author,
The quality of the manuscript has been significantly improved. However, one minor revision is recommended: in addition to my latest comment in the preivios review, it is advisable to show if there is any influence of the gas pressure on the sensor readings.

Author Response

Dear Reviewer,

Thank you for the swift feedback regarding the manuscript. As in the last response letter, we will respond to the points mentioned in the review report and indicate which changes have been made to manuscript in accordance with those points. Quoted text will be in italic font; text added to the manuscript will be placed in red (and highlighted in the PDF-version of the manuscript).

Dear Author,

The quality of the manuscript has been significantly improved. However, one minor revision is recommended: in addition to my latest comment in the preivios review, it is advisable to show if there is any influence of the gas pressure on the sensor.

Response: Thank you for the very positive assessment. The overall influence of the pressure on the gas sensor depends on the used absorption line as well as the method of the data analysis. For an ideal gas, increasing the pressure increases the amount of molecules in the gas in the same fashion, such that an increase in the absorption signal is to be expected. However, at the same time the increased pressure increases the linewidth due to collision broadening, such that the peak level barely changes. For this sensor concept, we only investigated the peak level as a method to determine the concentration, but it is absolutely possible to use a fitting procedure and calculate “the area under the curve”, such that an increased pressure would directly relate to a bigger value. We added the following lines to the manuscript to include this information in an abbreviated form.

Increasing the pressure leads to an increased amount of absorbers in a given volume, and hence a stronger absorption; however, the increased pressure also results in stronger collision broadening, leading only to slight changes when comparing the maximum absorption level to the baseline. Operating the device at higher pressure can be beneficial when using different methods, e.g., integrative measurements or fitting routines, to determine the concentration.

We want to thank the reviewer once more for the swift comments to improve the manuscript and hope we have incorporated the suggestions adequately.